# A Fast Thermal 1D Model to Study Aerospace Material Response Behaviors in Uncontrolled Atmospheric Entries

**DOI:** 10.3390/ma15041505

**Published:** 2022-02-17

**Authors:** Serena R. M. Pirrone, Camilla Agabiti, Adam S. Pagan, Georg Herdrich

**Affiliations:** 1Bioinspired Soft Robotics Laboratory, Istituto Italiano di Tecnologia (IIT), Via Morego 30, 16163 Genova, Italy; 2The BioRobotics Institute, Scuola Superiore Sant’Anna, Viale Rinaldo Piaggio 34, 56025 Pontedera, Italy; camilla.agabiti@santannapisa.it; 3Institut für Raumfahrtsysteme/Institute of Space Systems, Pfaffenwaldring 29, 70569 Stuttgart, Germany; pagan@irs.uni-stuttgart.de

**Keywords:** aerospace materials, demise behavior, uncontrolled atmospheric entry, ablation model

## Abstract

A preliminary thermal 1D numerical model for studying the demise behavior of stainless steel 316L, silicon carbide (SiC) and carbon fiber reinforced polymer (CFRP) during uncontrolled atmospheric entry is proposed. Test case modeling results are compared to experimental data obtained in the framework of ESA Clean Space initiative: material samples were exposed to different heat flux conditions using the Plasma Wind Tunnel (PWT) facilities at the Institute of Space Systems (IRS) of the University of Stuttgart. This numerical model approximates the heating history of the selected materials by simulating their thermal response and temperature profiles, which have trends similar to the experimental curves that are found. Moreover, when high heat flux conditions are considered, the model simulates the materials’ mass loss due to the ablation process: at the end of the simulation, the difference between the experimental and the modeled results is about 17% for CFRP and 35% for stainless steel. To reduce the model’s uncertainties, the following analysis suggests the need to consider the influence of adequate material thermophysical properties and the physical-chemical processes that affect the samples’ temperature profile and mass loss.

## 1. Introduction

The Design for Demise (D4D), as promoted, e.g., via the European Space Agency’s (ESA) Clean Space initiative [1], is gaining attention as one of many approaches towards reducing the amount of debris that pollute the Low Earth Orbit (LEO) environment. Indeed, expended launcher stages, decommissioned satellites and exploded or collided spacecraft have created a large amount of space debris leading to an increased risk for future space missions [2,3,4]. Recent studies have analyzed the evolution of the amount of space debris, which is ever growing [2,5,6,7,8] and will continue to do so if no effective mitigation strategies are adopted [2].

During re-entry, the space debris can break-up in small fragments that may impact the ground if they do not burn up completely [4]. Performing a controlled re-entry (i.e., the spacecraft undergoes an actively performed de-orbit maneuver to impact within an uninhabited area) is not always an option due to the increased costs and issues of technical reliability associated therewith [4,9]. Therefore, a passive approach is arguably preferable, wherein the spacecraft’s constituent components burn up to a degree that the ground risk emanating from the sum of residual debris items is considered non-critical [4,10,11].

Among the factors affecting the ‘demisability’ of a space vehicle during an uncontrolled atmospheric entry from the Low Earth Orbit (LEO), the material choice is one of the most significant steps in the design of these vehicles [10,12,13,14]. Past studies carried out in the last decade were aimed at improving the understanding of aerospace materials’ characteristics with the occurrence of ablation mechanisms taking place when exposed to high thermal loads. 

When metallic materials are subjected to high thermal loads, they absorb heat, leading to an important increase in temperature, which may rise up to a critical value (i.e., the melting temperature) at which the material starts melting. Even though melting is the main process characterizing metal demise, the oxidation represents an additional important process during which a superficial film is created [15]. As observed, e.g., for stainless steel AISI316L, the following occurs [15,16]:Melting: As indicated in Figure 1, this endothermic process takes place after the bulk material temperature reaches its melting point as a result of the high thermal loads occurring during atmospheric entry, absorbing most of the excess net heat;Oxidation: The front surface oxidizes due to the interaction between the oxidizing, partially dissociated species in the boundary layer and the material, with the consequent growth of a superficial passive (i.e., persisting) oxide layer. The exothermic nature of this process implies limited additional heat release on the surface.

When ceramic materials are subjected to high thermal loads, the main processes potentially resulting in mass loss are sublimation and/or melt and in the case of non-oxide ceramics, such as SiC, oxidation, which can transition between a passive and active phenomenology [17,18]. Silicon carbide (SiC) specifically undergoes to the following processes when exposed to heating through an air plasma [17,18]: Sublimation: As a potential surface ablation process indicated in Figure 2, sublimation occurs only at temperatures in excess of around 3000 K, a point that is rarely attained in typical destructive atmospheric entry scenarios from the Low Earth Orbit (LEO);Oxidation: Under most relevant high-enthalpy air flow conditions, SiC undergoes passive oxidation with some analogy to metals, such as stainless steel. At high temperatures and low pressures, which are borderline achievable under uncontrolled LEO entry conditions, the formation of volatile oxide occurs, and the material starts losing mass as it transitions into active oxidation.

Regarding the complex ablation process of composite materials, several studies have been conducted.

At moderate temperatures, organic composite materials, such as carbon fiber reinforced polymers (CFRP), undergo a volume ablation process that leads to the formation of char [19], thus deteriorating mainly due to the pyrolysis process of its matrix material, which is often an epoxy resin. The resin degrades into a mixture of gases that are driven out of the porous char residue. The material may swell far beyond its initial volume because of the increased internal pressure, consequently creating an insulating structure that causes a reduction in the internal heating, potentially effectively slowing the ablation rate over time. During the resin’s evaporation, the remaining char burns as a consequence of the surface oxidation producing carbon monoxide.

Many pertinent insights can be gained from studies of the behavior of a carbon/epoxy composite laminates exposed to fire, as, e.g., conducted by Tranchard et al. [20] using a 3D thermochemical model to study the dynamic interdependencies of the various ablation phenomena and material properties governing heat and mass transfer. McKinnon et al. [21] analyzed the thermal degradation of carbon fiber composites caused by a fire event on the basis of their thermophysical properties and key chemical reaction parameters and noted, e.g., that the thermal conductivity observed in the plane of the material was around 15 times higher than the value achieved in depth, that internal mass transport could be inhibited due at high material laminate densities, and that surface oxidation rates are insignificant at temperatures corresponding to incident heat fluxes below $60\frac{\mathrm{kW}}{m^{2}}$.

Fritsche [22] carried out a detailed review of the ablation processes that occur in carbon fiber reinforced polymers (CFRP) when exposed to heating conditions of relevance to destructive atmospheric entry, subsequently describing a 1D model accounting for the major physical and chemical processes to represent and quantify thermo-ablative CFRP behavior occurring during uncontrolled atmospheric entry in a phenomenologically meaningful and computationally efficient manner.

When CFRP is exposed to high thermal loads, transformation processes take place, beginning at its surface and gradually shifting towards its inner layers, consequentially leading to a significant change in its mechanical characteristics. The ablation process includes both pyrolysis and oxidation. Both these processes depend on a number of factors outlined in the following:Pyrolysis: As indicated in Figure 3, the epoxy matrix close to the front surface (i.e., the sample’s surface is located at *x = x_f_* and directly exposed to the plasma flow, whereas the back surface is the sample’s surface located at *x = x_b_*, which is not directly exposed to the plasma flow) is decomposed into a mixture of gaseous species as products of the endothermic chemical process of pyrolysis. The decomposed CFRP layer extends its volume into the virgin material. Following this decomposition, a porous carbon char remains. The char layer is the region through which the pyrolysis gaseous species escape into the boundary layer flow [22].Oxidation: The carbon char actively oxidizes in its superficial layer primarily due to the impinging, partially dissociated oxygen [22]. This chemical process is exothermic, and it results in gaseous products that are dissipated, differently from the surface oxidation concerning the metallic materials.

In this context, different modeling approaches were proposed to describe the ablation process of CFRP when exposed to high thermal loads. Among these, Quintiere et al. [19] considered a one-step first order decomposition model based on Equation (1):(1)$$\frac{d\alpha }{dt}=\frac{1-\alpha }{1-\mu }\;k\left(T\right)$$
where $\alpha =\frac{m-m_{i}}{m_{f}-m_{i}}=\frac{\frac{m}{m_{i\;}}-1}{\mu -1}$ represents the mass loss rate; $k=a_{P}\exp\left(-\frac{E_{a}}{R\;T}\right)$ is the Arrhenius rate; $E_{a}$ is the decomposition’s activation energy; $a_{P}$ is the pre-exponential factor; *m* is the mass; the subscripts *i* and *f* are, respectively, the initial and final conditions; and *μ* is the mass residue fraction. Equation (1) is solved for the mass ratio $\frac{m}{m_{i}}$. In their analyses, Quintiere et al. [19] considered both thermal conductivity and specific heat as functions of temperature, simulating a model where there is no need to know the properties of each constituent component material.

A different CFRP ablation model was proposed by Tranchard et al. [20], who considered the decomposition mechanism as a two-step process described by Equation (2):(2)$$\begin{array}{cccc} \rho \;c_{P}\frac{\partial T}{\partial t}= & \;\nabla \;\left(\;\Lambda \;\nabla T\;\right)- & \left[h-\;h_{g}\right]\frac{d\rho }{dt}- & \left[\;\begin{array}{c} m_{gx}\;\\ m_{gy}\;\\ m_{gz}\; \end{array}\right]\nabla h_{g}\\ ⏟ & ⏟ & ⏟ & ⏟\\ \left(a\right) & \left(b\right) & \left(c\right) & \left(d\right) \end{array}$$
where (*a*) is the heat transfer; (*b*) is the anisotropic heat conduction; (*c*) is the composite decomposition; and (*d*) is the gas transport that is linked to the internal pressure. Along with Equation (2), Tranchard et al. [20] considered a set of additional equations to characterize the boundary conditions, which include an equation comprising the heat fluxes acting at the surface directly impacted by the heat source and an equation related to the heat fluxes concerning the surface that remains unexposed. Equation (2) was solved in function of the mass evolution, the time to ignition and the temperature profile along the material sample, and the results were compared to experimental results, thus indicating the model’s capability to predict the material’s behavior. In relation to the CFRP ablation, McKinnon et al. [21] assumed the thermal degradation of the carbon fiber composite to be described by a four consecutive chemical reactions mechanism formed by first-order reactions, except for the last one, which is a second-order reaction. They observed a gradual increase in the sample’s thickness during the laboratory test, in accordance with the observations made by Quintiere et al. [19], who found a 100% increase in the original thickness value at the end of the simulation. The four-step reaction mechanism, presenting an error as low as 7% on average when compared to experimental findings, may be considered a suitable scheme to model the mass loss process of the carbon fiber composite due to thermal degradation. 

Whereas McKinnon et al. [21] and Quintiere et al. [19] modeled the variation of the sample’s thickness, Fritsche [22] did not consider this aspect, instead focusing on formulating an inexpensive model primarily representing pyrolysis- and oxidation-related thermo-ablative processes, as reproduced hereafter in a summarizing fashion.

The local heating in the material due to incoming heat flux ${\dot{q}}_{h}$ is described by Equation (3):(3)$$\frac{d{\dot{q}}_{h}\;}{dx}=\rho \;c_{p}\frac{\partial T}{\partial t}$$
where *ρ*, *c_p_* and *T* are the, respectively, local values of density, specific heat at constant pressure and temperature. 

The surfaces at temperature *T* re-radiate the heat flux to the outside as:(4)$${\dot{q}}_{r}=\varepsilon \;\sigma \;T^{4}$$
where *σ* is the Stefan–Boltzmann constant and ε is the emissivity of the material.

Between the internal surfaces, the heat that is exchanged through a conductive mechanism can be expressed as follows:(5)$${\dot{q}}_{c}=\lambda \frac{\partial T}{\partial x}$$
where λ is the material thermal conductivity. The pyrolysis process is an endothermic process that adds a negative contribution to the heat fluxes balance and decreases the density of the resin $\rho_{p}$ as:(6)$${\dot{\rho }}_{p}=-F\;\rho_{p\;}$$
where *F* is the pyrolysis reaction rate. When the gaseous volatile compounds diffuse in the charred region from internal to external layers of the CFRP material, the following heat flux is exchanged between layers:(7)$$\;{\dot{q}}_{g}={\dot{m}}_{g}\;c_{p,g}\frac{\partial T}{\partial x}$$
where ${\dot{m}}_{g}$ is the mass flux and $c_{p,g}$ represents the specific heat at constant pressure of the pyrolysis gas mixture. After the decomposition of the resin, a carbon char layer remains, and it is subjected to oxidation due to the impinging oxygen. This process is exothermic and the oxidation heat flux ${\dot{q}}_{\mathrm{ox}}$ is expressed as follows:(8)$${\dot{q}}_{\mathrm{ox}}={\dot{m}}_{\mathrm{ox}}\;h_{\mathrm{ox}}$$
where $h_{\mathrm{ox}}$ is the heat of oxidation and ${\dot{m}}_{\mathrm{ox}}$ is the rate governing the recession of the front layer by oxidation.

Fritsche considers (a) a discretization in space, where the spatial derivatives that are in the differential equations have to be approximated by spatial differences, and (b) a discretization in time where, for each time step, first the temperature is computed and then the heat fluxes are computed. Therefore, the temperature distribution at a given time is used as known input data to calculate the corresponding heat flux. In the numerical algorithm, Fritsche [22] considered the 1D wall divided in N layers and applied this scheme for a sample of 20 mm thickness divided in 10 layers, which are characterized by a constant thickness. The multi-step calculation is executed for each layer in order to obtain the temperature profile along the CFRP sample thickness accounting for the contribution of conduction, radiation, pyrolysis and oxidation processes. Considering the different approaches highlighted above, the work developed by Fritsche [22] appears to be the most adequate for modeling the CFRP ablation in our study because it represents an adequate compromise between accuracy in the representation of the ablation phenomena and relative computational efficiency. 

Concerning the metallic and ceramic materials, their ablation behavior can be well represented with an approach that is similar to the procedure reported above for the CFRP. However, metals and ceramics demise processes do not involve the pyrolysis process. Therefore, the incoming heat contribution (${\dot{q}}_{h}$) is different and can be expressed as follows:(9)$${\dot{q}}_{h}={\dot{q}}_{\mathrm{aero}}+{\dot{q}}_{\mathrm{ox}}-{\dot{q}}_{\mathrm{rad}}$$

In relation to metal ablation behavior, it has to be considered that, if the local heating exceeds the melting temperature, the material starts melting and receding in thickness at a rate described by the following expression: (10)$$\dot{s}=\frac{{\dot{q}}_{h}}{\rho \;h_{\mathrm{fus}}}$$
where $h_{\mathrm{fus}}$ is the material’s heat of fusion.

Besides through mathematical modeling, the ablation behavior shown by the aerospace materials when exposed to high thermal loads has been investigated also experimentally. In particular, beginning with the ESA TRP Characterisation of Demisable Materials conducted in the context of ESA’s Clean Space initiative, multiple material characterization test campaigns have been carried out at the Institute of Space Systems (IRS) of the University of Stuttgart on various material types using the IRS Plasma Wind Tunnel (PWT) entry test facilities PWK1 and PWK4 [23,24], the results of which have contributed to ESA’s ESTIMATE demisable material database [25]. By exposing a material sample to a high-enthalpy air flow within a vacuum vessel, boundary layer heating conditions relevant to uncontrolled atmospheric entries are created around the stagnation point of the plasma probe in which the material sample is positioned, with the effected conditions having been previously characterized by calorimetry and Pitot pressure measurements. A functional schematic of an IRS PWT test facility is depicted in Figure 4, with the investigated materials and effected test conditions for the presently discussed demisable materials test campaign described in further detail in [24].

The primary objective of the Characterisation of Demisable Materials PWT test campaigns was to assess the phenomenological and quantitative destructive and non-destructive responses of various aerospace materials subjected to atmospheric entry conditions to provide a database through which ground risk mitigation techniques and predictive methodologies could be improved and verified [24].

During each experiment, the material’s response was monitored throughout exposure via pyrometry at both the exposed front and the back surface of the sample, optical emission spectroscopy in the stagnation point region as well as HD video recordings of the demise process [24]. In addition, the physical properties of the samples were recorded pre- and post-test, and virgin as well as previously exposed samples were subjected to a dedicated characterization of total and measurement-device specific, temperature-dependent emissivities in the IRS Emissivity Measurement Facility (EMF) [26]. Selected high-temperature materials were further subjected to non-destructive steady-state thermal response testing in the PWT facilities to provide data for the extraction of quantitative catalytic properties.

In this work, a numerical model for studying the demise behavior of these materials is developed and the modeling results are compared to those obtained experimentally at IRS: the numerical model is a thermal 1D model that considers the major chemical and physical processes occurring during atmospheric uncontrolled entry in the following materials: CFRP, stainless steel AISI316L and SiC. Moreover, the mass losses characterizing the CFRP and the stainless steel AISI316L are estimated.

## 2. Materials and Methods

This chapter introduces a thermal 1D model that was developed to estimate the trends and parameters that were found experimentally at IRS [24].

### 2.1. Investigation on the Materials’ Temperature Profiles

Our numerical model aims to calculate the temperature profiles over time and along the thickness of metallic, ceramic and composite material samples when subjected to atmospheric uncontrolled entry conditions from LEO regions by simulating their thermal response (i.e., the temperature profiles during the test time *t* and along the thickness *x* of materials samples). The developed numerical model was a thermal 1D model and considered a space and time discretization of the sample. The simulated sample’s thickness was divided in N = 15 layers of constant thickness and a constant simulation step of 1 s was set. In the following, the governing equations used for describing the ablation behavior of each material are proposed.

Case-study 1: Carbon Fiber Reinforced Polymer (CFRP)

In this case-study, the numerical model was based on Equation (11), which considered the heat stored in the material, the heat due to conduction, the pyrolysis contribution and the heat exchanged between pyrolysis gases and charred region within this region, respectively [20]:(11)$$\rho c_{P}\left(\frac{\partial T}{\partial t}\right)=\lambda \frac{\partial }{\partial x}\left(\frac{\partial T}{\partial x}\right)-F\rho_{P}h_{P}-\rho_{g}Flc_{P,g}\left(\frac{\partial T}{\partial x}\right)$$

To solve Equation (11), the following data were needed:*ρ*, *c_p_*, *λ*, which represent the density, the specific heat at constant pressure and the thermal conductivity of the CFRP sample, respectively. During the simulation, these parameters varied with the temperature *T* and their values were calculated using data measured by the Österreichisches Gießereiinstitut in the framework of the DLR-led CHARDEM activity for CFRP (XN-60-60S/CP0031), which is available on ESA’s ESTIMATE materials database [25].The pyrolysis process was considered as a one-step chemical process and, therefore, the reaction rate F was computed using Equation (12):(12)$$F=A\;exp\left(-\frac{E_{P}}{RT_{P}}\right)$$
where *A* is the pre-exponential factor; *E_P_* is the activation energy of pyrolysis process; *R* is the universal gas constant; and *T_P_* is the activation temperature of the pyrolysis process (this temperature was about 600 K). The term *ρ*_P_ represents the matrix’ momentary density changing as it pyrolyzes, which can in approximation be empirically coupled to the temperature as shown in Table 1. The term $h_{P}$ is the enthalpy of the pyrolysis process.$\rho_{g}$ represents the density of the gas mixture produced by the pyrolysis, which is here assumed as consisting primarily of a mixture of carbon monoxide (CO) and carbon dioxide (CO_2_).The term l is the thickness of each of N layers comprising the sample. The parameter $c_{P,\;g}$ is the specific heat at constant pressure of the gas mixture produced by the pyrolysis.

Equation (11) is solved for a set of adequate boundaries and starting conditions applied to the sample’s front and back surface. The heat fluxes at the boundary region are described in Equation (13), where the aerothermal heat flux, the conductive heat flux, the radiative heat flux, the heat flux due to the oxidation of the carbon char and the heat flux exchanged between pyrolysis gas mixture and charred regions are considered [20]: (13)$${\dot{q}}_{\mathrm{BC}}={\dot{q}}_{\mathrm{aero}}-\lambda \frac{\partial T}{\partial x}-\sigma \varepsilon T^{4}+\rho_{C}lh_{\mathrm{ox}}k-h_{g}\rho_{g}Fl$$

The parameters used in Equation (13) were: ${\dot{q}}_{\mathrm{aero}}$ is the aerothermal heat flux. The simulations were performed in two different heat flux conditions: (a) low heat fluxes conditions (the reference value used for the aerothermal heat flux was 410$\;\mathrm{kW}/m^{2}$), where the sample was subjected mainly to pyrolysis outgassing, and (b) high heat fluxes (the reference value used for the aerothermal heat flux was 1400 $\mathrm{kW}/m^{2}$), where both the pyrolysis of the epoxy matrix and the oxidation process of the char proceeded at high rates. Concerning the high heat flux condition case, the actual heat fluxes to which samples were exposed to during experiments differ from these reference values. This variation can be explained considering that the reference values of the heat fluxes are related to hemispherical probes, while flat probes were utilized during the experiments [24]. Assuming a fully or near continuous flow regime, the heat fluxes measured with flat and hemispherical head calorimeter probes can be correlated using Equation (14) [24]:(14)$${\dot{q}}_{\;\mathrm{flat}}=\frac{{\dot{q}}_{\;\mathrm{hemi}}}{\sqrt{2,3}}$$

In approximation, it was assumed that the measurements conducted using the cold-wall oxidized copper calorimeter probes correspond to the fully catalytic heat flux.

-*σ* =$\;5.670367\;10^{-8}\frac{W}{m^{2}K^{4}}$ is the Stefan–Boltzmann constant.-*ε* is the emissivity.-$h_{g}$ and $\rho_{g}$ are the enthalpy and the density of the gas mixture produced by the pyrolysis, respectively. -F is the pyrolysis reaction rate and it is described by Equation (12).-l is the thickness of each N-layer composing the sample. -$\rho_{c\;}$ is the char density.-$h_{\mathrm{ox}}$ is the heat due to the oxidation process.-k represents the reaction rate of the oxidation process and it is expressed as follows [27]:(15)$$k=\exp\left(-\frac{E\mathrm{ox}}{R\;T_{\mathrm{ox}}}\right)$$
where *E*_ox_ is the activation energy of the oxidation process; $T_{\mathrm{ox}}$ is the activation temperature of the oxidation process; and $T_{\mathrm{ox}}$ is about 1160 K.

The initial condition used in the model consisted of the initial value of *T* and it was considered constant along the sample’s thickness. The initial temperature *T*_0_ was assumed to be equal to the ambient temperature: *T*_0_ = 298 K. 

Case-study 2: Stainless Steel AISI316L 

In this case-study, the numerical model was based on the energy and mass balance equations reported below [28]. In particular, in Equation (17) the left term represents the rate at which the material loses mass once the melting temperature is reached, while the right term includes the heat fluxes that are responsible for the material ablation.
(16)$$\rho c_{p}\frac{\partial T}{\partial t}=\lambda \frac{\partial }{\partial x}\left(\frac{\partial T}{\partial x}\right)$$
(17)$$\frac{dm}{dt}=-\frac{A}{h_{\mathrm{melt}}}\left[{\dot{q}}_{\mathrm{aero}}-{\dot{q}}_{\mathrm{rad}}\right]$$

Equation (16) is valid as long as the temperature on the sample surface and in its interior remains below the material melting temperature. As in the previous case, the parameters *ρ*, *c_p_* and *λ* are functions of *T* and their expressions were obtained from stainless steel 316L data available on ESTIMATE [25]. The main parameters of Equation (17) were the following:-$\frac{dm}{dt}$ is the mass loss rate.-*A* represents the front surface of the material’s sample.-${\dot{q}}_{\mathrm{aero}}$ is the aerothermal heat flux. For metals, the simulations considered the heat flux conditions used in the previous case-study: low heat fluxes conditions (where the reference aerothermal heat flux value was 410$\;\mathrm{kW}/m^{2}$) and high heat fluxes (where the reference aerothermal heat flux value was 1400 $\mathrm{kW}/m^{2}$). In both heat flux condition cases, this analysis included a correction accounting for the different catalytic behavior of the sample material adopted in the experiments with respect to the metal simulated in our model. Indeed, the experiments considered copper samples, which show a catalytic efficiency that is significantly higher than that observed for stainless steel. Therefore, a reduction factor equal to 0.85 was assumed in our model in order to account for this relative difference. Moreover, concerning the high heat flux condition case, the correction factor due to the use of flat probes in the experiments instead of hemispherical probes was introduced.

Equation (16) was solved through the following boundary condition applied to the front surface and expressed in Equation (18):(18)$${\dot{q}}_{\mathrm{aero}}-\lambda \frac{\partial T}{\partial x}-\sigma \varepsilon T^{4}+h_{\mathrm{ox}}\rho_{\mathrm{ox}}K_{\mathrm{ox}}=0\;$$
where the conduction term $q_{\mathrm{cond}}$ and the heat flux due to the metal oxidation are significant. The oxidation was expressed through the parameter representing the heat of formation of the oxide layer ($h_{\mathrm{ox}}$), the oxide layer density ($\rho_{\mathrm{ox}}$) and the growth rate of the oxide layer thickness ($K_{\mathrm{ox}})$. The parameter $K_{\mathrm{ox}}$ was assumed to follow a logarithmic law, which describes the increasing thickness of the oxidized layer during the simulation time until a maximum value is reached. In detail, the logarithmic rate was typical of adherent and protective oxide layers (e.g., the oxide layer generated by the oxidation) and the maximum thickness was estimated from the experimental characterization of the material oxidation process occurring in air plasma conditions [15]. 

The thickness of the oxide layer increases gradually, and the oxidation heat is generated simultaneously. The sample’s front surface was exposed to the oxidation heat until the oxide layer reached the maximum estimated thickness: the growth rate of the oxide layer was formulated so that, when the maximum estimated thickness was achieved, the oxide layer was considered to be fully developed and the oxidation heat was set equal to zero, as shown in Figure 5. 

In Figure 5, it can be observed that the maximum value of oxidation heat is of the order of $10^{3}\frac{W}{m^{2}}$, which is much lower with respect to the aerothermal heat flux order ($10^{6}\frac{W}{m^{2}}$). In this condition, the emissivity observed on the sample’s front surface was assumed to be function of the temperature in accordance with experiments conducted at IRS, which showed the significant increase in the surface emissivity for material samples that were subjected to oxidation [24].

As in the previous case-study, the initial temperature *T*_0_ was assumed equal to the ambient temperature: *T*_0_ = 298 K.

Case-study 3: Silicon Carbide (SiC) 

In this case-study, the numerical model was based on the energy balance equation reported in Equation (19) [28]:(19)$$\rho c_{p}\frac{\partial T}{\partial t}=\lambda \frac{\partial }{\partial x}\left(\frac{\partial T}{\partial x}\right)$$

In Equation (19), the thermophysical parameters ($\rho ,\;c_{P},\;\lambda$) vary with *T*, in accordance with the experimental data given by the ESA’s database [25].

The energy balance in Equation (19) was solved using the following boundary conditions related the sample front surface:(20)$${\dot{q}}_{\mathrm{aero}}-\lambda \frac{\partial T}{\partial x}-\sigma \varepsilon T^{4}+h_{\mathrm{ox}}{\dot{m}}_{\mathrm{ox}}=0$$
where ${\dot{m}}_{OX}$ represents the ablation rate due to the oxidation process and it follows Arrhenius law. The oxidation of SiC was treated in a simplified way with respect to the complex phenomena that occur in the entry phase [17], in that the heat released by the oxidation was treated as a product of the reaction between oxygen and pure graphite [27]. In this case, the correction factors applied to the aerothermal heat flux followed the same rules of the ones introduced for stainless steel: in particular, a correction factor equal to 0.65 was applied to consider the different catalytic efficiency. 

The initial condition consisted of the initial temperature, which was set as in the previous cases to *T*_0_ = 298 K. 

### 2.2. Investigation on the Materials’ Mass Losses

One of the processes affecting the mechanical properties shown by the composite and the metallic materials during the atmospheric reentry was the mass loss.

This process was not observed in the SiC, due to the high resistance to degradation shown in the heat flux conditions that were considered [24]. In detail, the mass reduction observed in the CFRP was mainly due to the oxidation process of the carbon char, which took place when the composite material was exposed to temperature higher than 1160 K, which is the oxidation activation temperature. The stainless steel lost mass due to the melting process, which occurred at temperatures higher than 1644 K [29]. Considering that our numerical model discretized the samples in N = 15 layers, our findings relate the amount of composite and metallic materials that is lost in each N-layer due to the occurrence of the oxidation and melting processes, respectively. In the following, the mass loss estimation occurring in both materials was modeled considering the samples to be exposed to high heat flux conditions, which represent the more relevant mass loss case.

CFRP mass loss modeling

The oxidation process of the carbon char occurred when the temperature was higher than 1160 K. When the oxidation activation temperature was not reached, the thickness and mass of each sample’s layer i were simply described as follows, considering the thickness (s) and mass (m) of the whole sample:(21)$$s\left(i\right)=\frac{s}{N}$$
(22)$$m\left(i\right)=\frac{m}{N}$$

When the temperature was higher than 1160 K, the char oxidized inducing the sample mass reduction. The final mass of each sample’s layer i can be expressed as follows:(23)$$m_{\mathrm{final}}\left(i\right)=\left(\frac{m}{N}\right)+dm\left(i\right)$$

In Equation (24), dm(i) represents the mass loss of each layer i and can be described as follows:(24)dm(i)=ρ s(i) A
where *A* is the frontal surface of the sample and *ρ* is the density of each layer i. 

The parameter s(i) is the thickness of each layer i and it can be expressed as follows:(25)$$\left(i\right)=\frac{s}{N}-\frac{ds}{N}$$
where ds is the variation of thickness occurring in each layer due to the char oxidation. 

The thickness variation of each layer can be expressed in the following form:(26)$$ds=\frac{dm}{A\;\rho_{\mathrm{char}}}$$

In Equation (27), dm represents the mass reduction in each layer due to the char oxidation and can be expressed as follows:(27)$$dm=t_{\mathrm{test}}\;{\dot{m}}_{R}$$
where $t_{\mathrm{test}}$ is the simulation time and ${\dot{m}}_{R}$ represents the oxidation rate. 

The oxidation rate was computed considering a kinetic regime, which is an ablation regime where the chemical reaction rates dominate the oxidation process [27]. Therefore, ${\dot{m}}_{R}$ was modeled as follows:(28)$${\dot{m}}_{R}=14,801.5\;\exp\left(-\frac{184,220}{R\;T\left(i\right)}\right)\sqrt{p_{e}\;X_{O2}}$$
where T(i) is the temperature of each layer; $X_{O2}$ is the molar O_2_ fraction, and $p_{e}$ is the external flow pressure near the stagnation point.

Once the CFRP mass loss was computed, the blowing effect, which arises as the pyrolysis gases emanate from the front surface, thus modifying the flow field, was included in the study by considering a blowing factor, which is a parameter measuring the reduction in the effective aerothermal heat flux [22]. In particular, the blowing factor was computed as follows [22]: (29)$$\phi =\frac{\left(a\;B\right)}{e^{\left(a\;B\right)-1}\;}$$

In Equation (29), the parameters a, *B* are the following: -a is a constant depending on the gas, the flow condition and the Mach number; its value for CFRP is considered to be 1.3 [22].-$B=\frac{dm\;h_{0}-dm\;h_{w}}{{\dot{q}}_{\mathrm{aero}}}$, where dm is the mass variation between end and starting of the simulation; $h_{0}$ is the total aerodynamic enthalpy, and $h_{w}$ is the wall enthalpy.

Stainless Steel AISI316L mass loss modeling

The mass loss occurred when the melting temperature on the sample’s was exceeded at the front surface as well as internally. Once the melting temperature was reached, the rate at which the material’s sample loses mass was described by Equation (17). Similar to the CFRP case-study, the reduction in the stainless steel sample’s mass under exposure to high aerothermal heat flux was computed.

### 2.3. Numerical Implementation

In order to solve the energy balance equations, the calculations were conducted on MATLAB and the PDEPE (Partial Differential Equations for Parabolic and Elliptic problems) was adopted. The flow chart reported in Figure 6 summarizes how the numerical implementation works.

The PDEPE solver was suitable for solving partial differential equations that depend on one space variable *x* and time *t*. In particular, the PDEPE solves mathematical problems that are formulated as shown in Equation (30), which was solved with respect to a specific time interval *t*_0_
*< t < t*_1_ and a space interval *x_f_ < x < x_b_*:(30)$$c\left(x,t,T,\frac{\partial T}{\partial x}\right)\frac{\partial T}{\partial t}=x^{-m}\;\frac{\partial }{\partial x}\;\left(x^{m}\;f\left(x,t,T,\frac{\partial T}{\partial x}\;\right)\right)+s\left(x,t,T,\frac{\partial T}{\partial x}\;\right)$$

Our thermal model used the parameters *m*, *c*, *f*, *s* that are represented for each simulated material in Table 2.

Our model solved the governing equations exposed above by applying proper initial conditions (i.e., conditions at *t = t*_0_) and boundary conditions (i.e., conditions at the extreme sample’s surfaces *x = x_f_* and *x = x_b_*). In sum, at *t = t*_0_, the model computed the temperature distribution along the sample’s thickness (i.e., the solution was in the form *T(x*, *t*_0_*) = T*_0_(*x*)); at the extreme sample’s surfaces *x = x_f_* and *x = x_b_*, the model found the temperature concerning the boundaries *x = x_f_* and *x = x_b_* during the whole simulation time interval in which the simulation was run (i.e., *t*_0_
*< t < t*_1_). In particular, our model adopted boundary conditions that were expressed as follows: (31)$$a\left(x,t,T\right)+b\left(x,t\right)\;f\left(x,t,T,\frac{\partial T}{\partial x}\;\right)=0$$

At the boundary *x = x_f_*, the parameters *a*($x_{f}$) *=*$\;\;a_{f}$ and *b*($x_{f}$) *=*$\;b_{f}$ and are reported in Table 3.

When the boundary *x = x_b_* was analyzed, the numerical model considered two different cases: (a) a condition in which the sample is well insulated with respect to its support (i.e., adiabatic surface at *x = x_b_*) and (b) a condition when the surface experiences full radiative heat loss (i.e., radiative surface at *x = x_b_*). Therefore, the parameters *a*($x_{b}$)*=*$\;a_{b}$ and *b*($x_{b}$)=$\;b_{b}$ that were used for the selected materials were the following: *a_b_ =* 0 and *b_b_ =* −1 in the adiabatic case; and *a_b_ =* −*σ ε*$\;T^{4}$ and *b_b_ =* −1 in the radiative case.

## 3. Results

The following section shows the temperature profiles and the mass losses estimated with our numerical model and their comparison with the results found experimentally at IRS [24]. 

### 3.1. Materials’ Temperature Profiles

In all the graphs, the temperature profiles were modeled assuming a radiative and an adiabatic condition on the back surface (cases represented with green and blue curves, respectively), as explained above; the temperature profiles obtained experimentally are reported in red.

Case-study 1: Carbon Fiber Reinforced Polymer (CFRP)

Under the low heat flux condition, the results shown in Figure 7 were obtained.

In both the cases where adiabatic and radiative conditions on the back surface were considered, the experimental temperature peak reached on the front surface was lower than the one obtained with our numerical model. When a radiative back surface was considered, (a) the experimental temperature peak reached on the back surface was higher than the modeling result; (b) the temperature gradient observed in the model was higher than the one observed experimentally; and (c) the growth rate and trend of the temperature profiles were similar on the front surface and different on the back surface. When an adiabatic back surface was assumed, (a) the experimental temperature peak on the back surface was lower than the modeling result; (b) the modeled temperature gradient between front and back surfaces was lower than the difference observed experimentally, and (c) the temperature profiles showed similar trend and growth rate on both the front and back surfaces.

Under the high heat flux condition, the results displayed in Figure 8 were achieved. In both the cases in which adiabatic and radiative conditions on the back surfaces were assumed, (a) the experimental temperature peak reached on the front surface was lower than the one predicted with our numerical model, and (b) the experimental temperature gradient between front and back surfaces was lower than the model’s finding. When a radiative back surface was considered, (a) the experimental temperature peak reached on the back surface was higher than the model result, and (b) the growth rate and trend of the temperature profiles were similar on the front surface and different on the back surface. When an adiabatic back surface was assumed, (a) the experimental temperature peak on the back surface was lower than the modeling result, and (b) the temperature profiles showed similar trend and growth rates on both the front and the back surfaces.

Case-study 2: Stainless Steel AISI316L

When a low heat flux condition was considered, Figure 9 shows the obtained results.

In this analysis, the oxidation process was included, and it involved the material surface exposed to the heat flux by affecting its optical properties (i.e., the emissivity). In the assumption of radiative back surface, (a) the modeling steady-state temperature was lower than the experimental case; (b) the temperature gradient observed in the model was lower than the experimental one; and (c) there was a good matching between model and experimental results in terms of growth rate and trend of the temperature profiles. Instead, in the case of adiabatic back surface condition, (a) the steady-state temperature given by the model was very close to the value of the front surface temperature reached experimentally; (b) the modeling temperature gradient was lower than the experimental one, and (c) also in this case the temperature profiles showed similar growth rate and trend.

Due to the single-color pyrometric measurement of both the front and back surface temperature, it is considered that the near-black-body assumption for the back surface measurements by which the experimentally determined temperature curves shown in this study are obtained is inaccurate for highly reflective surfaces, such as unoxidized or lightly oxidized stainless steel. Thus, pending an emissivity correction, the given experimental back surface temperatures merely constitute an extreme lower boundary of the actual temperature. This is not a concern for SiC or CFRP, which themselves feature considerably higher emissivities almost approaching unity.

In a high heat flux condition, front surface oxidation is relevant also in this case. The results are shown in Figure 10. When the back surface was assumed to be radiative, (a) the experimental temperature peak was higher than the modeling one; (b) the modeling temperature gradient was lower than the one observed experimentally; and (c) the temperature profiles showed similar growth rates and trended up to 180 s, time in which the material sample started to melt. In the assumption of adiabatic back surface, (a) the modelled steady-state temperature was higher than the experimental one; (b) the experimental temperature gradient was higher than the modelled value; and (c) the temperature profiles growth rates and trends were similar up to 180 s in both the experimental and model case.

Case-study 3: Silicon Carbide (SiC)

The ablation behavior of SiC was not simulated in the case of low heat flux condition because the material demonstrates a high resistance to demise and, therefore, such a test was not conducted. The results related to the high heat flux condition are displayed in Figure 11. Under the assumption of radiative back surface, the modeling and the experimental temperature profiles exhibited a good agreement in terms of temperature gradient, steady-state temperature, growth rate and trend. When the back surface was considered adiabatic, (a) the modeling temperature peak was higher than the experimental one; (b) the experimental temperature gradient was slightly higher than the modeling results; and (c) the temperature profiles showed similar trend and growth rate.

### 3.2. Materials’ Mass Losses

Following the procedure reported in Section 2.2., the reduction in mass related to the CFRP sample when subjected to high heat flux conditions is shown in Figure 12.

Figure 12 shows that the first 10 layers are completely ablated by the first 150 s, whereas the other 5 layers remain intact and with constant mass. The total mass lost by the end of the simulation was about 67% of its initial value (i.e., 4.2 g): this result appears realistic, although it is slightly underestimated with respect to the mass loss observed at the end of the experiments, where the post-test mass results to be 0.68 g, corresponding to a mass loss of almost 84% of the initial mass. On the basis of the results shown in Figure 12, the blowing factor was computed following the procedure reported in Section 2.2. and amounted to 0.96, indicating a small but significant effect of outgassing.

The sample mass loss observed for stainless steel AISI316L when subjected to high heat flux conditions is shown in Figure 13. In this case, the sample’s mass remained constant within the first 30 s; then, the material sample underwent a steep mass decrease and this condition took place within the time range in which the sample reached the melting temperature.

The total mass loss by the first 30 s of exposition to high heat flux conditions was about 67% of its initial value: this result cannot be directly compared to the experimental characterization since the testing procedure at IRS resulted in a termination of the experiment after melt was observed at around the 185 s mark, but before the sample had fully demised.

## 4. Conclusions

This work presented a thermal 1D model that approximates the heating history of the considered materials by simulating their thermal response when subjected to uncontrolled atmospheric re-entry. In detail, we considered materials exposed to heat flux conditions of 410 $\mathrm{kW}/m^{2}$ and 1400$\;\mathrm{kW}/m^{2}$. 

Our major findings are the temperature profiles observable on the material front and back surfaces during the simulation time and the amount of mass lost at the end of the simulation. In this context, our results were compared to the values obtained experimentally at IRS [24], and a good consistence was found. The comparison of the modeling vs. experimental results was performed with reference to (a) the highest temperature value reached at the end of the modeling simulation, (b) the shape of the temperature profiles of material surfaces during the entire simulation, (c) the growth rate and trend of the temperature with respect to the simulation’s duration, and (d) the mass loss characterizing the materials at the end of the test.

However, due to the complexity of the chemical and physical processes that occur in the materials during uncontrolled atmospheric entry conditions, several simplifying assumptions were adopted in our model setting and they affected the degree of uncertainty of our model’s performance. Among these factors, developing a numerical model independently on the duration of both the pyrolysis and oxidation processes significantly affected the overall uncertainty of our model simulations for all the analyzed materials and heat flux conditions. Regarding the modeling of CFRP’s ablation behavior, relevant uncertainty sources lie in the omission of the delamination process (i.e., the material is simplified as isotropic) and the internal transpiration cooling from the pyrolysis process. These factors affect the results, leading to higher simulated surface temperatures and a slightly underestimated sample mass loss with respect to the experimental findings. Concerning the ablation behavior modeling of metallic and ceramic materials, neglecting the variation of structural properties and chemical composition and the influence of the oxidation process on the material mechanical properties represents a simplifying assumption that explains the differences found between modeling and experimental findings. 

Therefore, a further improvement of our thermal 1D model would consist of involving a set of equations that include all the neglected aspects mentioned above, in order to reproduce with a higher consistency the temperature profiles and mass losses observed in the materials during the experiments and therefore achieve a better understanding of their ablation behavior.

## Figures and Tables

**Figure 1 materials-15-01505-f001:**
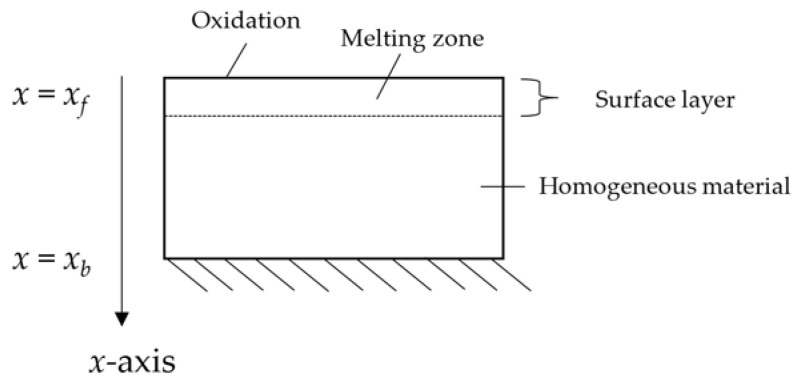
Demise-relevant processes occurring in stainless steel AISI316L.

**Figure 2 materials-15-01505-f002:**
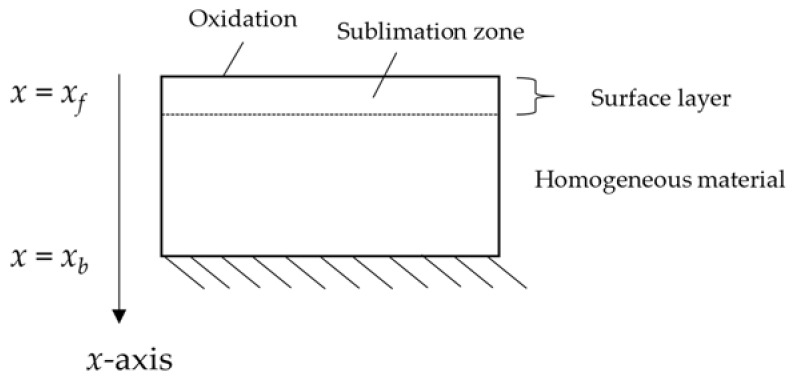
Relevant ablation processes occurring in SiC.

**Figure 3 materials-15-01505-f003:**
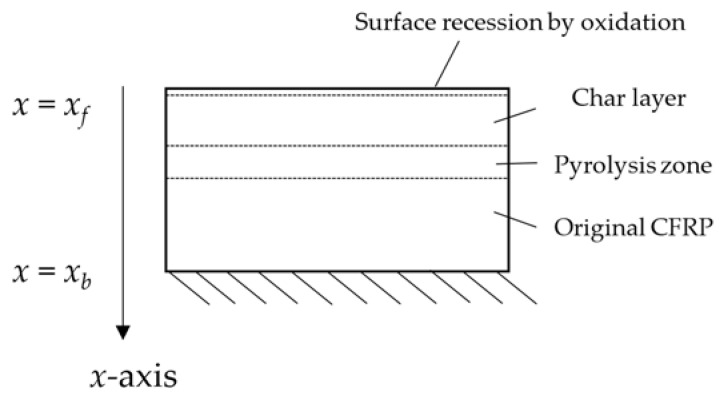
Ablation processes occurring in CFRP.

**Figure 4 materials-15-01505-f004:**
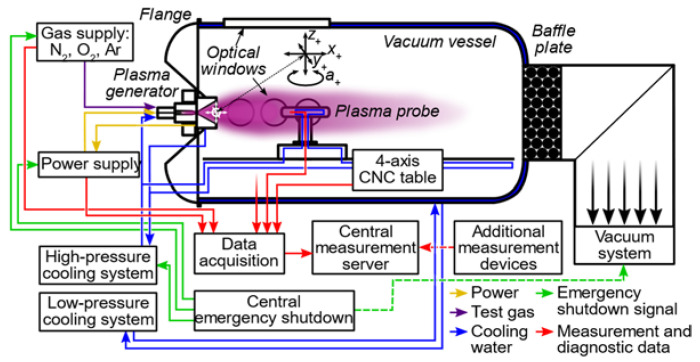
Functional schematics of IRS Plasma Wind Tunnel test facility in operation.

**Figure 5 materials-15-01505-f005:**
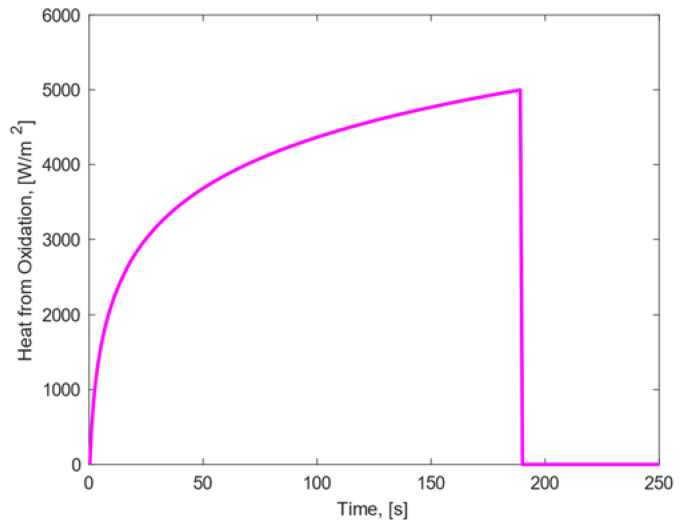
Heat release by surface oxidation of stainless steel 316L.

**Figure 6 materials-15-01505-f006:**
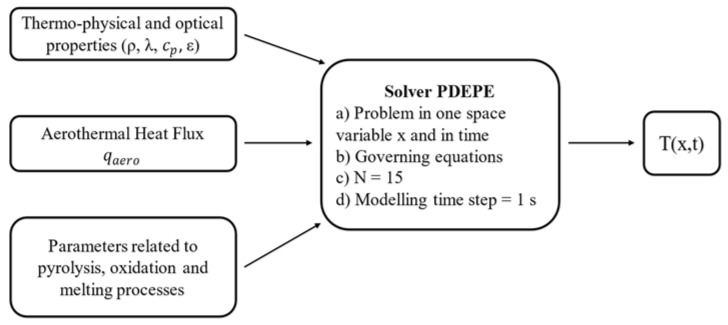
Numerical model implementation flow chart.

**Figure 7 materials-15-01505-f007:**
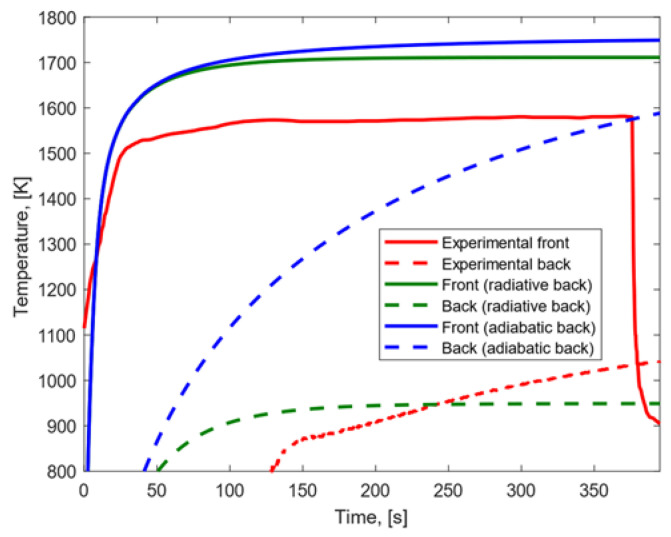
Experimental and modeling temperature profiles for CFRP subjected to low heat flux.

**Figure 8 materials-15-01505-f008:**
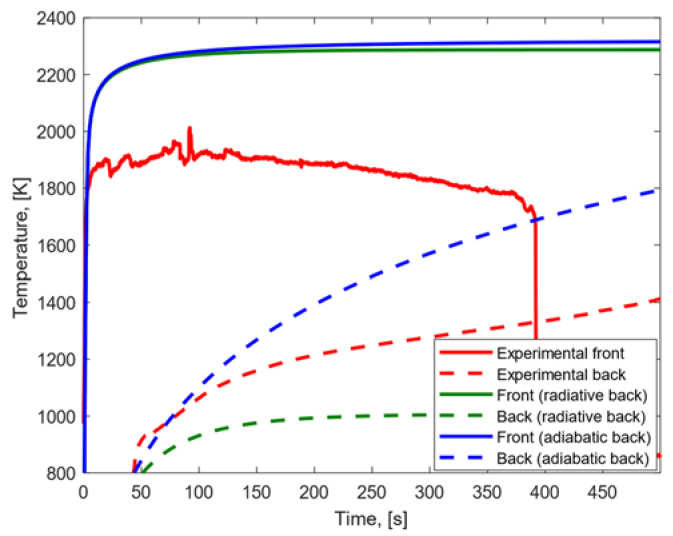
Experimental and modeling temperature profiles for CFRP subjected to high heat flux.

**Figure 9 materials-15-01505-f009:**
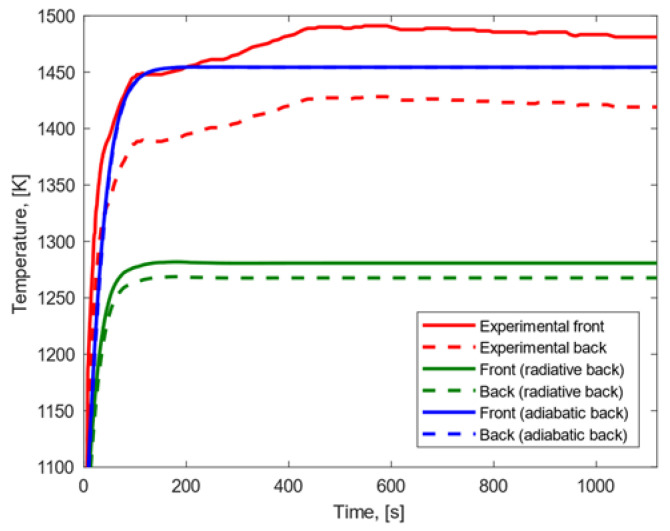
Experimental and modeling temperature profiles for stainless steel AISI316L subjected to low heat flux.

**Figure 10 materials-15-01505-f010:**
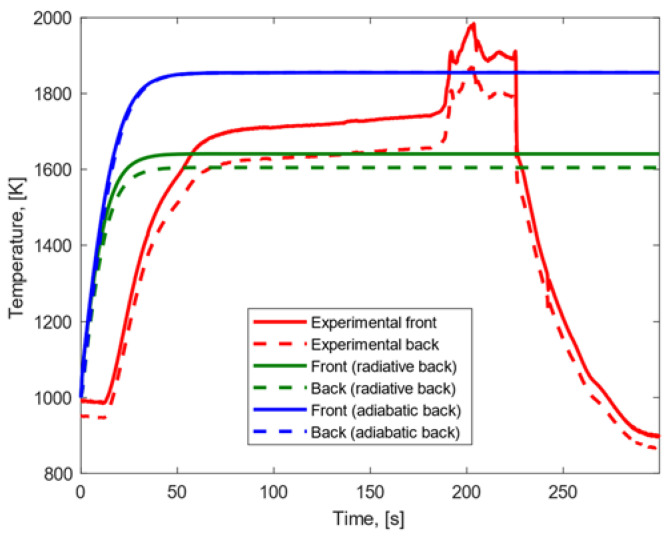
Experimental and modeling temperature profiles for stainless steel AISI316L subjected to high heat flux.

**Figure 11 materials-15-01505-f011:**
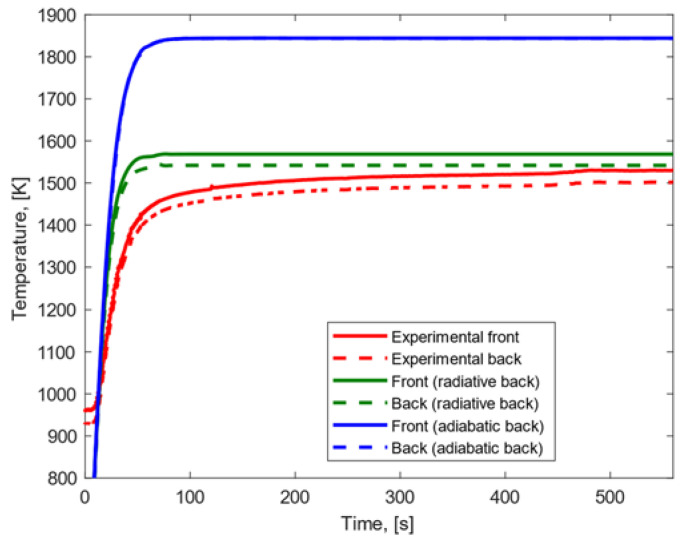
Experimental and modeling temperature profiles for silicon carbide subjected to high heat flux.

**Figure 12 materials-15-01505-f012:**
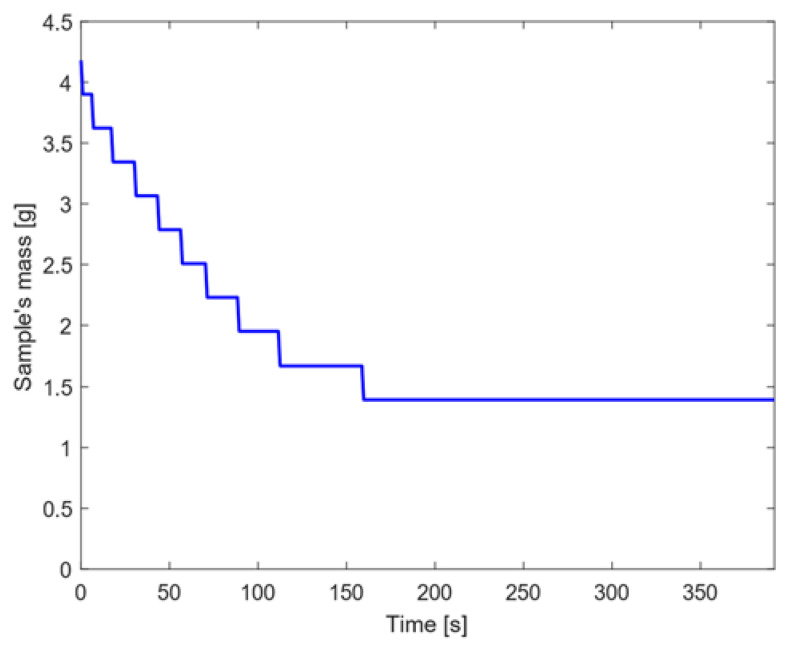
Sample’s mass decrease for CFRP when high heat flux is considered.

**Figure 13 materials-15-01505-f013:**
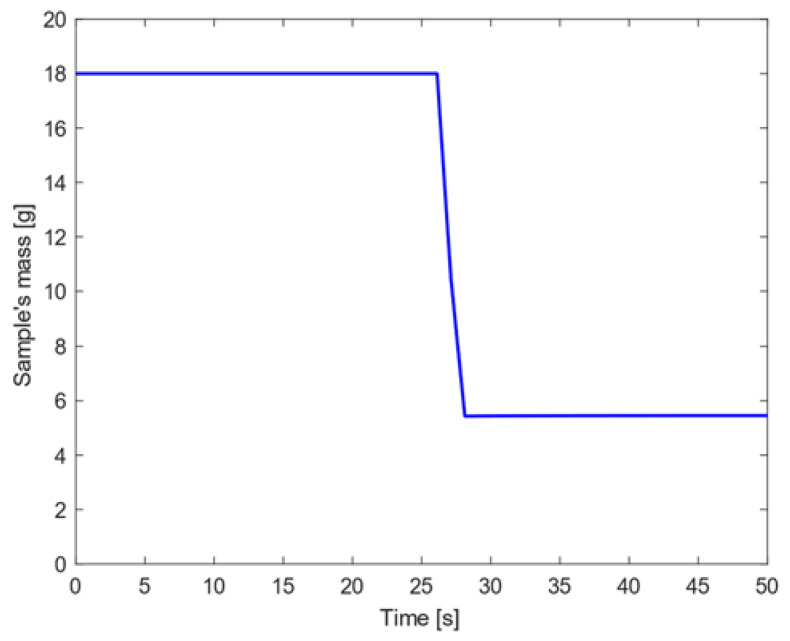
Sample’s mass decrease for stainless steel AISI316L when high heat flux is considered.

**Table 1 materials-15-01505-t001:** Matrix density of CFRP as a function of the pyrolysis temperature [25].

*T* < *T_p_*	*T* > *T_p_*
$\rho_{P}$ = 1170 $\frac{\mathrm{kg}}{m^{3}}$	$\rho_{P}=-0.0003\;T^{2}+0.3592\;T+1047.5$

**Table 2 materials-15-01505-t002:** Parameters *m*, *c*, *f*, *s* for CFRP, stainless steel 316L and SiC.

CFRP	Stainless Steel 316L	SiC
*m =* 0	*m =* 0	*m =* 0
$c=\rho \;c_{P}$	$c=\rho \;c_{P}$	$c=\rho \;c_{P}$
$f=\lambda \frac{\partial T}{\partial x}$	$f=\lambda \frac{\partial T}{\partial x}$	$f=\lambda \frac{\partial T}{\partial x}$
$s=\rho_{P}\;h_{P}\;F-\rho_{g}\;b\;F\;c_{p,g}\frac{\partial T}{\partial x}$	s=0	s=0

**Table 3 materials-15-01505-t003:** PDE parameters a and b on the front surface of CFRP, stainless steel 316L and SiC.

CFRP	Stainless Steel 316L	SiC
$a_{f}$ $\;\;{\dot{q}}_{\mathrm{aero}}\;$ $+{\dot{q}}_{\mathrm{ox}}$ $-{\dot{q}}_{\mathrm{rad}\;}$ $-{\dot{q}}_{\mathrm{pyr}}$	$a_{f}$ $\;\;{\dot{q}}_{\mathrm{aero}}\;$ $+{\dot{q}}_{\mathrm{ox}}\;$ $-{\dot{q}}_{\mathrm{rad}}$	$a_{f}$ $\;\;{\dot{q}}_{\mathrm{aero}}\;$ $+{\dot{\;q}}_{\mathrm{ox}\;}$ $-{\dot{q}}_{\mathrm{rad}}$
$b_{f}=1$	$b_{f}=1$	$b_{f}=1$

## Data Availability

The experimental data upon which this study is based are available via the European Space Agency’s online “European Space maTerIal deMisability dATabasE” (ESTIMATE) [25], access to which can be requested through the ESA Space Debris Office.

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
