# Peer review of "A Fast Thermal 1D Model to Study Aerospace Material Response Behaviors in Uncontrolled Atmospheric Entries"

_materials, 2022, doi:10.3390/ma15041505_

Round 1

Reviewer 1 Report

The article contains the interesting problem of study Aerospace materials response behaviours in uncontrolled atmospheric entries. The following problems were noted in the article:

  1. Research related to material modeling covers three different materials and the authors strive to develop only simple 1-D models. In the discussion, even according to the authors, too much simplifications were made during the modeling process and the models must be refined because they do not fully reflect the phenomena that occur. The research should therefore be considered incomplete.
  2. The article does not present conclusions and does not analyze in a countable way the differences between the results of model and experimental studies
  3. The research on the state of art includes 21 items, 5 of which are authors' positions and will not adequately reflect the state of the art in the world.
  4. Lost references (p. 6)
  5. The results presented in the figures from 6 to the end are not properly analyzed and the descriptions do not include the essence of comparing the simulation results and their verification.

Reviewer 2 Report

General remarks

The aim of the paper was to develop a 1-D thermal model that considers the major chemical and physical processes occurring during atmospheric uncontrolled entry in some materials: CFRP, Stainless Steel AISI316L and SiC and to compare the results of the numerical simulation with the experimental results obtained for the same materials at Institute of Space Systems (IRS) of the University of Stuttgart.

Two of the authors have some recent contributions in the last 5 years in this field:

  1. Loehle, S.; Zander, F.; Eberhart, M.; Hermann, T.; Meindl, A.; Massuti-Ballester, B.; Leiser, D.; Hufgard, F.; Pagan, A.S.; Herdrich, G., et al. Assessment of high enthalpy flow conditions for re-entry aerothermodynamics in the plasma wind tunnel facilities at IRS. CEAS Space Journal 2021, 10.1007/s12567-021-00396-y, doi:10.1007/s12567-021-00396-y.
  2. Anih, S.; Pagan, A.S.; Koch, H.; Martinez, P.; Laufera, R.; Herdrich, G. Investigation of long-duration crewed space missions solid waste management using waste for energy and volume recovery (WEVR) experiments.
  3. Anih, S.; Pagan, A.S.; Koch, H.; Martinez, P.; Laufera, R.; Herdrich, G. Waste for Energy and Volume Recovery (WEVR) using inductively heated plasma generator.
  4. Massuti-Ballester, B.; Pagan, A.S.; Herdrich, G. Oxidation and heterogeneous catalysis on titanium Ti-6Al-4V in high-enthalpy flows.
  5. Galla, D.A.; Herdrich, G.; Komurasaki, K.; Massuti-Ballester, B.; Momozawa, A.; Pagan, A.S.; Soga, R. Investigation of passive to active oxidation transition on ultra high temperature ceramics.
  6. Mione, M.; Massuti-Ballester, B.; Pagan, A.S.; Herdrich, G. Water-cooled adjustable material probe design for the evaluation of transient heat fluxes of high temperature materials. pp. 927-931.
  7. Pagan, A.S.; Massuti-Ballester, B.; Herdrich, G. Experimental thermal response and demisability investigations on five aerospace structure materials under simulated destructive re-entry conditions.
  8. Loehle, S.; Fasoulas, S.; Herdrich, G.; Hermann, T.; Massuti-Ballester, B.; Meindl, A.; Pagan, A.S.; Zander, F. The plasma wind tunnels at the institute of space systems: Current status and challenges. pp. 1-15.

I found some similarities in the first chapter with two other papers, cited in the paper. The title of the first paper is "Modelling Behaviour of a Carbon Epoxy Composite Exposed to Fire: Part II-Comparison with Experimental Results" (paper number 9 from the Reference list) and was also published in Materials journal. The title of the second paper is “Pyrolysis model for a carbon fiber/epoxy structural aerospace composite” (paper number 10 from the Reference list) and was published in a Sage journal: Journal of Fire Sciences.

For example, the text from lines 84 to 88 is almost identical to the text presented in the MDPI paper.

 “Tranchard et al. [9] developed a 3-D thermochemical model to predict the temperature profile, the mass loss and the decomposition of a carbon/epoxy composite laminate when exposed to a strong and sustained fire event. This model is based on the energy balance equation that takes in account the energy stored in the material, the heat conduction, the thermal decomposition, the gas mass flow in the composite and the internal pressure.”

Another example: the text from lines 95 to 100 is almost identical to the text presented in the Sage journal.

“appear rotated 45 degrees with respect to the previous layer. Therefore, a composite with these characteristics can be considered a material with quasi-isotropic thermal transport properties through the plane of the material. McKinnon et al. [10] found that the thermal conductivity in the plane of the composite was higher than the value obtained in-depth (by approximately a factor of 15) and they observed that a) the mass transport was inhibited due to the high density of the laminae in the composite and b) the oxidation rate is not relevant at the temperature produced by heat fluxes up to”

Please change (rephrase) this text!!!

In the references there are 5 titles (from 21) of the authors of the paper. I recommend reducing this number.

Some suggestions and questions:

  1. The authors determine the values for the density, the specific heat and the thermal conductivity of the CFRP sample for a specific CFRP material: XN-60-60S/CP0031. For the proposal thermal model do they have to determine this data for each material?
  2. How did the authors determine the activation energy of pyrolysis process (Ep) and how did the authors determine the relation presented in Table 1, of the temperature dependent value of matrix density?
  3. Did the authors consider the gas density not temperature dependend? It is correct?
  4. Did the authors estimate the number of the entry values in the 1-D thermal model and how can a researcher find all this values?
  5. Why did the authors choose to discretize the samples in N = 15 layers? How did they select this number?
  6. I recommend transforming Scheme 1 in a Figure (Figure 6, for example).
  7. I saw in the Results chapter that, for CFRP material, even though in the experimental cases there are serious differences between adiabatic front and radiative back, the numerical model presents close results for them. How can the authors explain that?
  8. In the abstract, the authors mentioned that “when high heat flux conditions are considered, the model simulates a material’s mass loss due to the ablation process: at the end of the simulation the difference between the experimental and the modelled results is about 17% for CFRP and 35% for stainless steel”. Did the authors consider this difference acceptable? In my opinion, a 35% difference it is a little too high.
  9. Line 239-240 there is a formatting error: “L'origine riferimento non è stata trovata”

Reviewer 3 Report

Ref.comments to the paper titled as “A fast thermal 1-D model to study aerospace materials response  behaviors in uncontrolled atmospheric entries” written by the authors: Serena R. M. Pirrone, Camilla Agabiti† , Adam S. Pagan and Georg Herdrich.

Currently, due to the painstaking study of the cosmos, including not only the celestial bodies themselves, but also the apparatuses constructed on Earth, there is a need to study the behavior of the materials under various dynamic conditions. From this point of view the study and explanation the process of the thermal influence on the steel, silicon carbide and carbon fiber reinforced polymer is very important. From this point of view the current paper is modern and actual.

It should be firstly remarked that the author has made a literary search consisting of 21 references, but so little part of them have been written during last 3 years. Indeed, the authors has the knowledge of the problem and can find the ways to solve it, but they have to extend the references list and add 5-7 papers yet written on last 3-5 years.

Introduction part is good and don’t contradict with the general knowledge in this area. Among the different ablation process explanation the authors have chosen some good approach.

Section Materials and Methods is interesting as well. Would you please to explain how the materials losses after the treatment can be coincided with the refractive index change?

Results part presents a good coinciding between experimental results and simulation ones.

Discussion part looks as a Conclusion one. Presented summarized results have not contradicted to our physical knowledge.

So, the article is good. But, in my local opinion, this paper can be published in the Journal after the corrections mentioned above.

Round 2

Reviewer 1 Report

The article is a very introductory approach to modeling aerospace materials response behaviours in uncontrolled atmospheric entries

Even in the opinion of the authors "the goal was to develop a preliminary study".

Both the methodology itself is only a preliminary approximation and laboratory experiments do not allow for statistical and scientific confirmation of the model's results. They only indicate the legitimacy of conducting research in this direction and the chance to develop a correct model in the future.

The planned experiment to confirm the correctness of the simulation model does not quantify the modeling results, and what the authors also confirm, is only a single result and in no way justifies the results of the numerical model. The observed large differences in the results confirm only the insufficient form of the model and the imperfections of the methodology.

Author Response

Dear Reviewer,

thank you for providing very useful suggestions. In the methodology as well as in the results and conclusions sections we have also highlighted that our study was a preliminary assessment to evaluate the importance of the major processes that occur when the materials are exposed to high and low heat fluxes. This will certainly be the basis for improved design of future experimental and modeling research activities on this very important topic.

Best regards,

Serena Pirrone 

Reviewer 2 Report

The paper can be published as it is.

Author Response

Dear Reviewer,

Thank you for your very useful suggestions provided to improve the quality of out manuscript. All your suggestions were addressed accordingly.

Best regards,

Serena Pirrone